# Digital mental health service engagement changes during Covid-19 in children and young people across the UK: Presenting concerns, service activity, and access by gender, ethnicity, and deprivation

Duleeka Knipe[1], Santiago de Ossorno Garcia[2,3], Louisa Salhi[2,4]*, Nimrah Afzal[2], Samaryah Sammut[2], Lily Mainstone-Cotton[2], Aaron Sefi[2,5], Amanda Marchant[6], Ann John[6]

1 Population Health Sciences, Bristol Medical School, University of Bristol, Bristol, United Kingdom, 2 Kooth Plc, 3 Department de Psicologia, Universidad Alfonso X el Sabio, Villanueva de la Canada, Madrid, Spain, 4 School of Environment, Education and Development, University of Manchester, Manchester, United Kingdom, 5 Exeter University, Exeter, United Kingdom, 6 Population Data Science, Swansea University, Swansea, United Kingdom

* lsalhi@kooth.com

## Abstract

The adoption of digital health technologies accelerated during Covid-19, with concerns over the equity of access due to digital exclusion. The aim of this study was to assess whether service access and presenting concerns differed before and during the pandemic. Sociodemographic characteristics (gender, ethnicity, and deprivation level) were examined to identify disparities in service use. To do this we utilised routinely collected service data from a text-based online mental health service for children and young people. A total of 61221 service users consented to sharing their data which represented half of the service population. We used interrupted time-series models to assess whether there was a change in the level and rate of service use during the Covid-19 pandemic (April 2020-April 2021) compared to pre-pandemic trends (June 2019-March 2020) and whether this varied by sociodemographic characteristics. The majority of users identified as female (74%) and White (80%), with an age range between 13 and 20 years of age. There was evidence of a sudden increase (13%) in service access at the start of the pandemic (RR 1.13 95% CI 1.02, 1.25), followed by a reduced rate (from 25% to 21%) of engagement during the pandemic compared to pre-pandemic trends (RR 0.97 95% CI 0.95,0.98). There was a sudden increase in almost all presenting issues apart from physical complaints. There was evidence of a step increase in the number of contacts for Black/African/Caribbean/Black British (38% increase; 95% CI: 1%-90%) and White ethnic groups (14% increase; 95% CI: 2%-27%), sudden increase in service use at the start of the pandemic for the most (58% increase; 95% CI: 1%-247%) and least (47% increase; 95% CI: 6%-204%) deprived areas. During the pandemic, contact rates decreased, and referral sources changed at the start. Findings on access and service activity align with other studies observing reduced service utilization.

**Data Availability Statement:** Data cannot be shared publicly because this is sensitive service data and the consent provided is for specific uses only. As researchers, we have to comply with the service's privacy policy. However, data may be made available from the Kooth for researchers who meet the criteria for access to confidential data by contacting research@kooth.com or lsalhi@kooth.com the corresponding author.

**Funding:** This project was supported by the Adolescent Mental Health Data Platform (ADP) funded by the MQ Mental Health Research Charity (MQBF/3 ADP) and Health and Care Research Wales (HCRW; SCF-22-10) and by the MRC and HDRUK through DATAMIND (MR/W014386/1).

**Competing interests:** Dr Louisa Salhi, Dr Nimrah Afzal, Samaryah Sammut and Aaron Sefi are currently employed by Kooth Digital Health and receive honorarium for their time. Dr Louisa Salhi also holds honorary researcher status at the University of Kent and the University of Manchester. Dr Nimrah Afzal holds an honorary research position at the University of Bath. Dr Santiago de Ossorno Garcia and Lily Mainstone-Cotton were previously employed by Kooth Digital Health at the time of data extraction and received honorarium for their time, but are no longer employed by Kooth Digital Health. Duleeka Knipe, Amanda Marchant and Ann John declare no conflict of interest, no remuneration was received from this work.

The lack of differences in deprivation levels and ethnicity at lockdown suggests exploring equity of access to the anonymous service. The study provides unique insights into changes in digital mental health use during Covid-19 in the UK.

## 1. Introduction

The Covid-19 pandemic has produced an unprecedented impact worldwide. Restrictive movement public health measures such as lockdown restrictions, social distancing and school closures, had to be taken by most counties in an attempt to minimise the spread of the outbreak and mitigate loss of life. While necessary, measures to manage the pandemic carried widespread indirect economic, social [1], and psychological consequences [2].

Reports have suggested that the loss of routine and social connection experienced by children and young people (CYP) has contributed to feelings of isolation and loneliness [3, 4]. CYP's declining emotional states as well as changes in their environments (such as having to study in shared spaces at home instead of school), have also been correlated with loss of concentration and ability to enjoy activities [5], negatively impacting their education outcomes, and their prospects in the labour market [6]. It is therefore clear that public health measures have disproportionately impacted the lives of CYP [4–8]. In fact, in England alone, it was predicted that 1.5 million CYP will need additional mental health support to combat the negative impacts of the pandemic [9].

In the UK, schools closed on March 23rd, 2020 [10], and remained so until the beginning of September, when new norms and regulations for learning environments were introduced. A systematic review of 36 studies concluded that school closures were associated with harm to the health and wellbeing of CYP across 11 countries [11]. A modest increase has been reported in CYP's anxiety, depression and traumatic stress, in comparison with previous population studies within the UK [12, 13]. This was a particular concern for CYP with existing vulnerabilities, including those with a history of mental/physical health problems or those from disadvantaged backgrounds [3]. Research suggests that vulnerable young people accessing digital mental health services (such as those at risk for adverse childhood experiences, those within a familial economic disadvantage, those who experience discrimination based on their gender, ethnicity or religion, and asylum seekers), are at a higher probability of heightened risk to mental health problems, loss of progress, as well as increased experiences of isolation [3, 14]. Thus, this is an important area of research as CYP with existing vulnerabilities would require additional psychological support [3].

One avenue for improving access to mental health support for CYP is utilising digital mental health support. There are numerous perceived benefits of online mental health interventions, such as anonymity, privacy, and emotional safety due to reduced emotional proximity to the practitioner, as well as increased flexibility, control, and accessibility to treatment [15–17]. Research exploring youth perspectives suggests that CYP value the potential for interactivity, personalisation, privacy, and sense of anonymity of online support [17–19]. The value of digital support was even more paramount during the Covid-19 pandemic, where digital platforms created opportunity for providing support amongst lockdown restrictions [20, 21], especially for CYP with existing vulnerabilities [3]. Digital mental health services, therefore, provided an avenue for providing accessible support during the Covid-19 pandemic.

Therefore, in an aim to combat the loss of services and provide the mental health support needed, many countries (such as the United Kingdom, the United States, Italy, Spain, China

and India, to name a few) created and / or significantly expanded their mental health digital support provision in the face of the pandemic [22]. While support was undeniably needed, as with any change, there were significant barriers and challenges tied to this transition (such as lack of privacy when accessing services from shared spaces) [23] with little known in terms of expected outcomes, utilisation and equitability across socio demographic groups.

It is important to consider the persistent inequalities highlighted in literature, in relation to access to mental health support. For example, research has suggested the CYP from ethnic minority backgrounds are more likely than others to experience issues relating to mental health access and treatment in the UK [24]. Additionally, it has been reported that individuals from ethnic minorities are less likely to access and engage with mental health services, creating a disparity in representation [25–27]. Gender disparities have also been reported in literature, with girls being more likely to recognise experiencing psychological problems and to seek help at mental health services, than boys [28, 29]. Conversely, literature suggests that digitalisation in the mental health sphere can have a positive impact on marginalised and traditionally underserved communities by reducing barriers to accessing care, reducing waiting times and travel costs [30]. Technology may therefore create an equitable route for mental health management, and help minimise disparity [31, 32]. This calls for an exploration of the impact of the pandemic on mental health service access and engagement among deprived communities.

While the digital shift has the potential to benefit underserved communities, research suggests that it may also contribute to the creation of a vulnerable 'digital underclass', i.e., individuals who choose to be internet non-users [33]. A variety of factors, such as lack of knowledge, financial deprivation, mental health difficulties and personal preferences have been associated with digital exclusion [34]. A recent review on digital mental health highlights the careful considerations to broaden access to mental support, by offering flexible and accessible forms of care. Different delivery options, such as video conferencing, telephone or text-based interventions and different platforms, such as web-platforms or mobile devices, may be well suited for different users' needs and priorities [35], such as anonymity concerns [36]. In aiming to reduce the digital divide and minimise disparity, it is crucial to explore issues relating to access to these services [37, 38], particularly in relation to the technological adoption resulting from the pandemic.

The Covid pandemic provided a unique situation that helped emphasise the importance of digital mental healthcare in times of crises [22]. It is crucial to consider how digital technologies were utilised during this time, by considering changes to referral sources, demographics of referred CYP, presenting issues, service use and patterns of access to mental health platforms. By identifying these changes, we can better understand the alternative or additional support needed by CYP as a result of the pandemic. Kooth (Kooth.com) is one such platform that CYP can access anonymously and immediately. The platform is free to the end users and is available regionally based on NHS commissioning in response to local needs. By examining the referral and access changes pre- and during Covid in relation to the Kooth platform, we will be able to inform and optimise future digital mental health services.

Established digital platforms provide a unique opportunity to examine how CYP access an online service for mental health support and how the service changed during the global crisis. We aimed to assess the impact of the pandemic on service use in terms of overall contact, referral source, type of contact, and the type of presenting issue (e.g. mental health, abuse) using interrupted time-series modelling, and explored whether the impact is different between socio-demographic groups (gender, ethnicity and deprivation level) in one digital mental health service for CYP operating in the UK. To evaluate the impact the study aimed to answer the following research questions:

i. Has there been a change in trends of contacts to the service pre- and during Covid-19 pandemic in the UK?

ii. Has there been a change in the referral source pre- and during Covid-19 pandemic in the UK?

iii. Has there been a change in service presenting concerns pre- and during Covid-19 pandemic in the UK?

iv. Is there any evidence that any changes observed vary by ethnic minority status, gender, and area-level deprivation pre and during Covid-19 pandemic in the UK?

## 2. Methods

### 2.1. Study setting

Kooth (Kooth.com) is an online text-based service for mental health and emotional wellbeing support service, funded by the UK National Health Service (NHS) and available for free to CYP aged 10–25 years in the UK. The service offers free, text-based, and anonymous support with no need for a referral, alongside an ecosystem of mental health support through moderated forum boards and curated materials with moderated activities (e.g., journaling, goal-setting, community forums). Young people can interact via text-message and receive support directly with practitioners through usually 60 minutes of synchronous text-based chats through the website. Users can also have asynchronous messages with practitioners to their inbox or indirectly engage with unstructured support and engagement through a community of peers in a moderated online forum. The platform and service provision has been previously identified as a positive virtual ecosystem through its theory of change [24]. The service regularly collects and monitors access, usage information, presenting concerns, and other service user information. It is however anonymous and the personal information about the user is limited as well as the capacity to verify service users' identity. The routinely collected data for this study was based on users who provided specific research consent to use their data within the platform.

### 2.2. Dataset

The datasets utilised in this study were routinely collected between 1st of June 2019 and the 31st of April 2021 at one digital mental health service (Kooth.com). The dataset contained only users who consent to use their data for research purposes at registration 50.25% of the total population of service users accessing during the period accessing Kooth.com.

At the time of this study Clinical Commissioning Groups (CCGs) were responsible for getting the best possible health outcomes for the local population/region in the UK. This involved assessing local needs, deciding priorities and buying services on behalf of their population. CCGs were responsible for commissioning healthcare including mental health services, urgent and emergency care, elective hospital services and community care (NHS Confederation, 2021). The UK National Health Service (NHS) is responsible for determining allocations of financial resources to CCGs. Total annual budgets given to CCGs cover the majority of NHS spending. A total of 34 regions structured by Clinical Commissioning Groups (CCG) with unchanged resource contracts were selected from a total of 97 (N = 5 decreased; N = 58 increased); the selection criteria of regions was used to prevent biases due to changes in resources that may affect the demand and capacity of the online service during the pandemic (resource increase or decrease during the observational period). The study selected only

regions of the service that had the same allocated number of resources before and during the pandemic, remaining therefore constant during the period.

Four datasets extracted from the service were linked and subsequently cleaned (e.g. removal of duplicates) for the analysis. The datasets contained unique user identifiers used for linkage to create one dataset containing information about access and engagement, presenting concerns during the period, user demographic information, region of access, and contact as service activity data, including logins, registration dates, and different service usage activities.

The Index of Multiple Deprivation (IMD) is a measure of relative deprivation for small areas (LSOAs). LSOAs comprise between 400–1200 households and are an aggregate of output areas that may be too small to maintain individual anonymity. The IMD was added to the dataset from the Ministry of Housing, Communities and Local Government public database [39], the CCG rank of multiple deprivation corresponding to each contract region from the service was added from the dataset, and the region of access (reported at service registration) was grouped and structured into CCGs for the service. Of note, not all contract areas were commissioned by CCG meaning that a trust could commission the service for partial CCG areas. Where only a partial match between IMD and Kooth could be obtained IMD data were excluded (29.4% of CCGs). No postcodes are collected in the service.

The service activity data was aggregated into practitioner-directed and self-directed interaction contacts to differentiate the online therapy provision of the service (asynchronous and synchronous chat) linked to human-to-human interaction and, online community engagement for its forums and other activities in which a human is not directly involved but will require some indirect interaction (albeit human moderation to approve content in the forums or asynchronous communication may take place). A combination of datasets and service information was uploaded to the Adolescent Data Platform (ADP) a UK secured e-research platform for analysis and collaboration, the datasets provided the group of variables used in the observational study.

## 2.3. Study variables

Service access contacts was our outcome of interest by month. We calculated a rate of contacts per user. To do this we identified the number of active users on the platform (i.e., denominator) by counting the number of users accessing the service at least once each month between June 2019 to April 2021. The numerator was the total number of contacts to the service during this period. The service activity is recorded in the platform engagement analytics. The service has different offerings with activities and different ways of engagement, users can engage in an online peer support forum, write in an emotional journal, request direct synchronous chats, or send asynchronous messages to practitioners. To assess whether the type of contact differed in the different periods, we categorized contacts into either practitioner-directed or self-directed. Practitioner-directed contacts were defined to be interactions that are human-mediated (interaction between a practitioner and user has taken place) like chats or therapeutic asynchronous messages, these are contacts for online therapy as opposed to online community forums [40]. Self-directed contacts were interactions with the platform through writing content in the online community forums, setting up goals, or using an emotional journaling tool. As a digital platform, users can still interact with the service through these kinds of self-directed interactions, despite human interaction may be required for safeguarding, safety, and moderation, it is not mediated by a human directly.

The digital service captures self-reported information about where users were referred from (educational settings; family and friends and word of mouth; primary or secondary care professionals; charitable and care organizations; and internet advertising, social media, and other

sources). It also has data on the presenting issue for which a user will engage with the service. Presenting concerns are practitioner-reported data about the problems, issues, or difficulties that users bring to the chat sessions, community forums and other activities in the service [41]. When a young person accesses the service their contact (if it leads to an interaction) is coded into a 'presenting issue'. A single contact can have several presenting issues. A total of 118 reported different presenting concerns were identified for the study. These were grouped into five high-level categories aligned with previous literature on the mental health impact of the pandemic (External, Physical health/other, Suicidality/self-harm/maladaptive coping, Mental health, Risk/abuse/safeguarding [S2 Table]). Consensus was agreed between the researchers on the groupings based on the literature and also ensuring that these groupings allowed for sufficient observations within each category to track trends over time.

In addition to the service engagement data above, we also utilized data collected on gender recorded at service registration ("*My gender is best described as*": i) male; ii) female; iii) Agender; iv) Gender-fluid). Gender identities were self-reported and reflect the options available to choose from at the time. We also collected data on ethnicity ("*My ethnicity most closely matches*": i) White; ii) Black/African/Caribbean/Black British; iii) Asian or Asian British; iv) Mixed/Multiple ethnic groups; and v) Not stated). Using the index of multiple deprivation of each CCG we assigned deprivation quintiles for each user based on the rank of their CCG [25]. We used the average IMD rank of each CCG location included (out of a possible 191) to calculate deprivation quintiles (higher ranks are least deprived) for each user in relevance to their area (partially disclosed at registration and group by CCG).

## 2.4. Analysis

For this analysis we define the time of the intervention (i.e., the start of the Covid-19 pandemic) as March 2020, this coincided with the date of the first lockdown period in the UK (23 March 2020). We provide summary statistics of the sample in terms of the number of active users per month during the pre-and pandemic periods, as well as the number of contacts by type and presenting concern. Trends of the rate of contact per user (overall and by type) are presented graphically with the intervention month marked in a red-dotted line. To examine the representativeness of our sample towards the user population at Kooth.com, Chi-square comparisons and effect sizes were performed to look at distribution differences in demographic characteristics between consented users and non-consenters and the magnitude of these differences between samples using effect size calculations.

An interrupted time-series analysis was used to assess whether there was evidence of a change in online service use during the pandemic period (April 2020—April 2021) compared to pre-pandemic (June 2019—March 2020) trends. We pre-specified the impact model for this analysis prior to data analysis. We examined the changes in the number of monthly contacts in the pre-pandemic and during the pandemic period. We fitted Poisson regression models with a scale parameter to account for overdispersion and the number of active users as the population offset. We fitted models for the overall number of contacts and then by type and presenting concern. We also fitted models for the number of new registrations by referral source, and further stratified the overall number of contacts by sex, ethnicity, and area deprivation level. We used the '*fp*' function in Stata statistical software (version 16.1) to incorporate longer-term time trends as fractional polynomials in all models as appropriate. We hypothesized that there would be both a step (i.e., level) and gradual (i.e., slope) change following the start of the pandemic in the number of contacts per user. For this, we included a binary coded variable in the model which represented the pandemic period (i.e., model step change), as well as an interaction term between time and intervention which models a slope change. Time periods were

defined as pre-pandemic (01.01.2018–31.03.2020) and during pandemic (01.04.2020–31.03.2021). All statistical analyses were conducted using Stata 16.

## 2.5. Ethics

The use of the service data and planned analysis was approved by the Ethics Committee of SWANSEA UNIVERSITY MEDICAL SCHOOL REC (*reference 2020–0050*). Ethical approval was granted on 8-10-2020. Subsequently, data extraction occurred on 19-05-21 of routinely collected service data which spanned back to June 2019 to April 2021. All the data was securely stored and shared using the ADP and the anonymity of users was ensured before the service shared any data with the researchers.

Importantly, consent for all the data extracted had been provided by users of the service, agreeing that their data to be used for future research and evaluation studies. Any users who did not consent to their data being shared were excluded from this data extraction and analysis.

There was no change in the routine care provided by the service and as such the data collected is being used following the terms and conditions of the service and in agreement with those service users who consented to their service data being used for this purpose.

## 3. Results

### 3.1. Routinely collected user's demographic characteristics

*Based on our sample of users from selected CCGs t*here were 61221 active consenting users during the study period with a roughly similar mean number of contacts per month in the pre- and during the pandemic period (Table 1).

From the sample of consented users composing the study 80.2% were White ($N$ = 49070) and 74.2% female ($N$ = 44412). Ages ranged from 13 to 20 years of age with an average of 16.1 ($SD$ = 1.96). From the sample extracted, 1003 participants identified as agender and 1732 as gender-fluid representing 3.4% of the total sample. Finally, almost 2% of participants did not state their ethnicity, with no difference in the level of missing by the pandemic period.

Differences could be present between user demographics who provided research consent which should be considered when interpreting the findings. Sample distributions between consenting and non-consenting users presented significant but very small differences in demographics with very small effect-sizes ranging v = [.06 - .07]. Males provided the highest frequency of research consent 61.1% relative to gender ($X^2$(4, 107670) = 542; p < .001) and those who did not state or self-reported their ethnicity with the lowest rate of consent rates 55.71% ($X^2$(3, 107670) = 142, p < .001) between consenters and non-consenters (S1A and S1B Table).

### 3.2. Change in service access

The number of users and contacts pre-pandemic and during the pandemic is shown in Table 1.

There was evidence of a positive pre-pandemic trend in the number of contacts per user, and that there was a step-change (i.e., sudden change) in the number of contacts at the start of the pandemic. When comparing trends in the periods, before the pandemic there was evidence of a 25% (95% CI: 9%-44%) increase rate in contacts per user per month (p = .001). There was a significant change in slope in March 2020, with an initial increase in contacts, followed by a subsequent decline which collectively resulted in an overall reduction in growth to a 21% (95% CI: 4%-41%) increase per month during the pandemic (p < .001; calculated as post-intervention slope = pre-intervention slope [1.25] * change in slope [0.97] = 1.21; Fig 1 and Table 1).

**Table 1. Descriptive table of users and contacts pre-pandemic and during pandemic.**

| | | COVID period* | |
|---|---|---|---|
| | | Pre- | During |
| Users | | | |
| Mean no. of active users per month (SD) | | 3579 (572.9) | 3792 (724.6) |
| Mean no. of active users per month by deprivation quintile (SD) | | | |
| | Q1—most deprived | 820 (117.6) | 864 (173.8) |
| | Q2 | 551 (109.6) | 506 (83.7) |
| | Q3 | 468 (87.3) | 577 (166.9) |
| | Q4 | 534 (168.4) | 555 (106.8) |
| | Q5—least deprived | 178 (51.3) | 148 (30.4) |
| | Missing | 1029 (148) | 1143 (207.5) |
| Contacts | | | |
| No. of contacts | | 189011 | 88687 |
| Mean no. of contacts per month (SD) | | 7000 (1215.7) | 7391 (1243.9) |
| Number of contacts n (%) | | | |
| | Article Comment | 6470 (3.4) | 1467 (1.7) |
| | Chat | 29490 (15.6) | 12442 (14) |
| | Forum Article Created | 9193 (4.9) | 5809 (6.6) |
| | Forum Comment | 13658 (7.2) | 9786 (11) |
| | Goal | 20652 (10.9) | 8957 (10.1) |
| | Journal | 62163 (32.9) | 31411 (35.4) |
| | Live Forum Comment | 1804 (1) | 400 (0.5) |
| | Message Sent | 45581 (24.1) | 18415 (20.8) |
| | Missing | 0 | 0 |
| Number of contacts by type n (%) | | | |
| | Practitioner-directed | 75071 (39.7) | 30857 (34.8) |
| | Self-directed | 113940 (60.3) | 57830 (65.2) |

\* Pre-pandemic (1st Jan 2018-31st Mar 2020); During (1st April 2020-31st Mar 2021)

We stratified contacts by type of activity (Table 2). There was a 33% (95% CI: 13%-57%) increase in self-directed contacts before the pandemic with no significant immediate change at the start of the pandemic. During the pandemic the increase in the rate of contacts reduced slightly to 28% (95%: CI 7%-54%) increase per month during the pandemic period.

For practitioner-directed contacts, there was no evidence of an increase in the number of contacts per user per month pre-pandemic, but evidence of a step-change increase (30%; 95% CI: 10%-53%) immediately following the start of the pandemic. There was no compelling evidence that the rate of change differed in the pandemic period compared to the period before.

## 3.3. Referral sources

There was clear evidence that the source of referrals to the service had experienced a step change during the pandemic period, with significant reductions in referrals via educational sources, and a rise in referrals via family/friend, other, care/social and internet sources (Table 3). However, there was no evidence for differences in the rate of the trends seen in relation to contacts with the digital service relating to referral sources changes in the pre- and during pandemic periods.

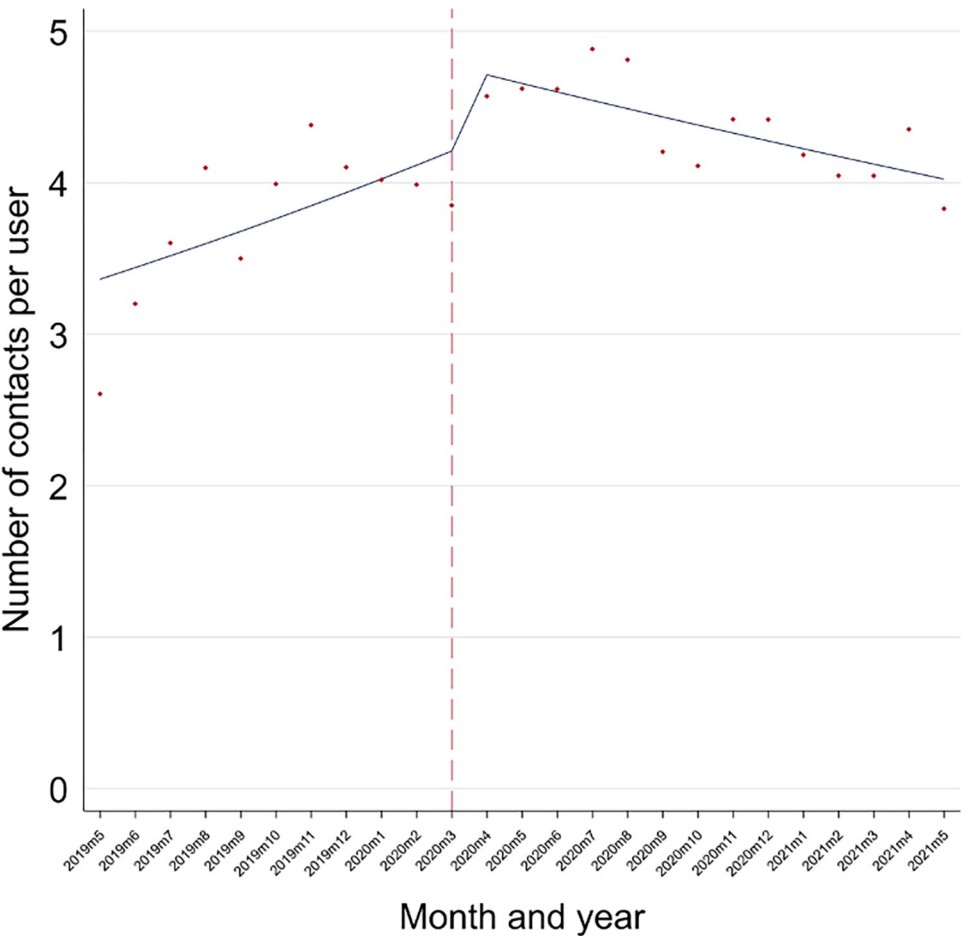

**Fig 1.** Time-series of the number of contacts per user pre- and during Covid-19 at the digital mental health service.

### 3.4. User's mental health presenting concerns

*Time series modelling by presenting concern is shown in Table 4. Note that each contact may have multiple presenting concerns recorded and as such trends do not mirror those for contacts per user overall shown above.* Prior to the pandemic there was no evidence that the trend line

**Table 2. Interrupted time-series model split by type of activity in the service.**

| Overall Model | | Rate ratio | 95% CI |
|---|---|---|---|
| | Pre-intervention slope | 1.25 | 1.09, 1.44 |
| | Change in intercept | 1.13 | 1.02, 1.25 |
| | Change in slope | 0.97 | 0.95, 0.98 |
| **Activity Type** | | | |
| Practitioner-directed | Pre-intervention slope | 1.33 | 1.13, 1.57 |
| | Change in intercept | 1.07 | 0.95, 1.21 |
| | Change in slope | 0.96 | 0.94, 0.98 |
| Self-directed | Pre-intervention slope | 1.07 | 0.86, 1.34 |
| | Change in intercept | 1.30 | 1.10, 1.53 |
| | Change in slope | 0.97 | 0.95, 1.00 |

**Table 3. Interrupted time-series model split by referral source self-reported in the digital service pre and during the pandemic.**

| Referral Source[1] | | Rate Ratio | 95% CI |
|---|---|---|---|
| Education | Pre-intervention slope | 1 | 0.88, 1.13 |
| | Change in intercept | 0.82 | 0.71, 0.94 |
| | Change in slope | 0.99 | 0.94, 1.04 |
| Family / Friends | Pre-intervention slope | 0.96 | 0.81, 1.13 |
| | Change in intercept | 1.38 | 1.18, 1.61 |
| | Change in slope | 1.01 | 0.95, 1.08 |
| Medical / Psych | Pre-intervention slope | 0.96 | 0.81, 1.15 |
| | Change in intercept | 1.13 | 0.94, 1.34 |
| | Change in slope | 1.02 | 0.95, 1.09 |
| Care / Social / Charity | Pre-intervention slope | 0.94 | 0.8, 1.11 |
| | Change in intercept | 1.26 | 1.06, 1.48 |
| | Change in slope | 0.98 | 0.92, 1.04 |
| Social media / Internet | Pre-intervention slope | 1.13 | 0.78, 1.62 |
| | Change in intercept | 1.67 | 1.2, 2.33 |
| | Change in slope | 0.91 | 0.8, 1.04 |
| Other | Pre-intervention slope | 0.92 | 0.69, 1.22 |
| | Change in intercept | 1.5 | 1.14, 1.96 |
| | Change in slope | 0.99 | 0.89, 1.1 |

[1] Self-reported referral source by the question "*How do you hear about us*?" at registration in Kooth.com

either increased or decreased (i.e., relative stable rate of contacts by presenting concern). Immediately after the pandemic, there was evidence of a step-change in the number of contacts per user for external (40% increase), mental health (81% increase), risk/abuse/safeguarding risk (100% increase), and suicidality/self-harm/maladaptive coping (57% increase). Following this initial increase in contacts a non-significant downward trend was resumed for all presenting concerns with the exception of suicidality/self-harm/maladaptive coping where

**Table 4. Interrupted time-series split by presenting concerns pre- and during-pandemic.**

| Presenting concerns | | Rate Ratio | 95% CI |
|---|---|---|---|
| External | Pre-intervention slope | 0.78 | 0.54, 1.13 |
| | Change in intercept | 1.4 | 1.05, 1.87 |
| | Change in slope | 1 | 0.96, 1.05 |
| Mental Health | Pre-intervention slope | 0.83 | 0.56, 1.23 |
| | Change in intercept | 1.81 | 1.36, 2.4 |
| | Change in slope | 0.99 | 0.95, 1.04 |
| Physical health / Other | Pre-intervention slope | 0.87 | 0.48, 1.56 |
| | Change in intercept | 1.5 | 0.96, 2.35 |
| | Change in slope | 0.97 | 0.9, 1.04 |
| Risk / Abuse / Safeguarding risk | Pre-intervention slope | 0.66 | 0.34, 1.27 |
| | Change in intercept | 2.03 | 1.24, 3.32 |
| | Change in slope | 1 | 0.93, 1.08 |
| Suicidality / Self-harm /Maladaptive coping | Pre-intervention slope | 0.93 | 0.64, 1.36 |
| | Change in intercept | 1.57 | 1.2, 2.06 |
| | Change in slope | 1.02 | 0.97, 1.06 |

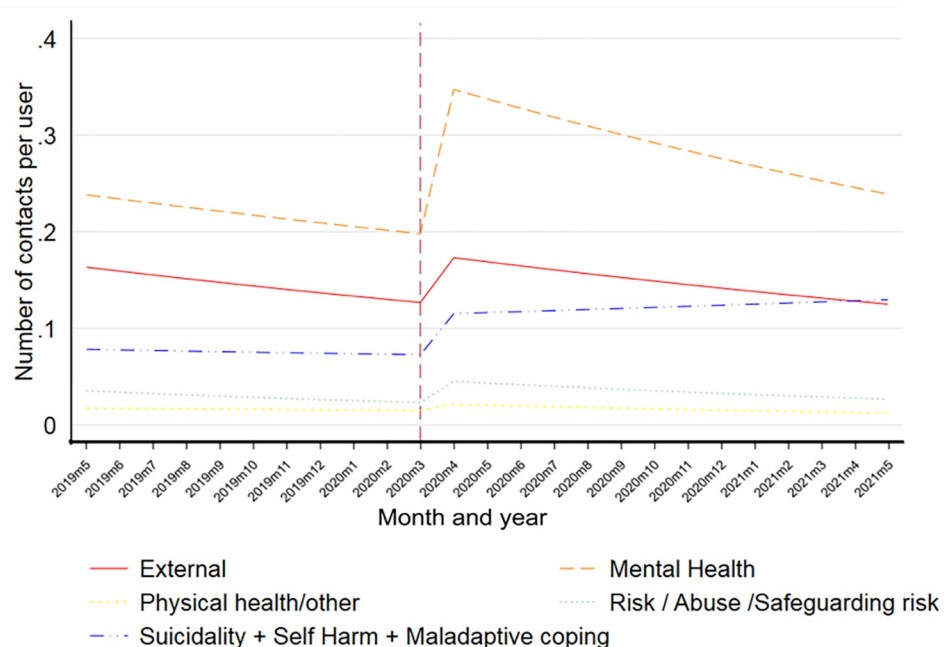

**Fig 2. Time-series of monthly presenting concerns and contacts pre- and during Covid-19 in the digital service.**

a slight upward trend can be seen, however this did not reach statistical significance. As the pandemic progressed there was no evidence that the rate of change during the pandemic (the slope of the trend line) differs from that before the pandemic (Fig 2).

### 3.5. Ethnicity, gender, and area deprivation level service access changes

We stratified the overall contacts per user by certain demographic factors (Table 5). There was evidence of a step increase in the number of contacts for Black/African/Caribbean/Black British (38% increase; 95% CI: 1%-90%) and White (14% increase; 95% CI: 2%-27%) ethnic groups. There was only evidence of a differing pandemic trend line (i.e., decreasing trend line) in the number of contacts per user for the Mixed/Multiple ethnic groups and the White majority group. When stratified by gender, we found evidence of a positive trend before the pandemic, followed by a step-change (15% increase; 95% CI: 3%-27%), and evidence that the trend line during the pandemic differed from the prior trend (i.e., decreasing trend) in females, but not other gender groups.

For the CCGs included for which we were able to derive an IMD rank there was no evidence that the slope changed in the pandemic period compared to the pre-pandemic period for any of the deprivation quintiles (Table 6).

There was evidence that the most (58% increase; 95% CI: 1%-247%) and least (47% increase; 95% CI: 6%-204%) deprived areas both experienced a step-change increase in the number of contacts per user when Covid-19 measures commenced and the impact of the pandemic started to be experienced in the UK.

## 4. Discussion

The research within this paper aimed to investigate the impact of the Covid-19 pandemic on service utilisation of Kooth (https://www.kooth.com/), a digital mental health platform for CYP. Changes in service use were examined before, during, and after the Covid-19 pandemic

**Table 5. Interrupted time-series for ethnicity and gender in the digital service pre- and during pandemic.**

| Ethnicity | | Rate Ratio | 95% CI |
|---|---|---|---|
| Asian /Asian British | Pre-intervention slope | 0.96 | 0.74, 1.24 |
| | Change in intercept | 1.13 | 0.93, 1.38 |
| | Change in slope | 1 | 0.97, 1.03 |
| Black / African/ Caribbean/ Black British | Pre-intervention slope | 0.91 | 0.6, 1.38 |
| | Change in intercept | 1.38 | 1.01, 1.9 |
| | Change in slope | 1 | 0.95, 1.05 |
| Mixed/Multiple ethnic groups | Pre-intervention slope | 1.4 | 1.11, 1.77 |
| | Change in intercept | 1.13 | 0.96, 1.34 |
| | Change in slope | 0.95 | 0.92, 0.98 |
| Not stated | Pre-intervention slope | 1.61 | 0.89, 2.93 |
| | Change in intercept | 1.02 | 0.68, 1.52 |
| | Change in slope | 0.98 | 0.91, 1.04 |
| White | Pre-intervention slope | 1.29 | 1.11, 1.49 |
| | Change in intercept | 1.14 | 1.02, 1.27 |
| | Change in slope | 0.96 | 0.94, 0.98 |
| **Gender[1]** | | | |
| Agender | Pre-intervention slope | 1.64 | 0.95, 2.83 |
| | Change in intercept | 0.81 | 0.54, 1.21 |
| | Change in slope | 0.97 | 0.91, 1.04 |
| Female | Pre-intervention slope | 1.31 | 1.13, 1.51 |
| | Change in intercept | 1.15 | 1.03, 1.27 |
| | Change in slope | 0.96 | 0.94, 0.98 |
| Gender Fluid | Pre-intervention slope | 1.3 | 0.78, 2.15 |
| | Change in intercept | 0.87 | 0.58, 1.3 |
| | Change in slope | 0.99 | 0.93, 1.05 |
| Male | Pre-intervention slope | 1.01 | 0.85, 1.2 |
| | Change in intercept | 1.08 | 0.93, 1.25 |
| | Change in slope | 0.99 | 0.97, 1.01 |

[1] The 'Not stated' option was excluded from the analysis due to low frequencies

onset specifically in terms of trends of contact, referral source, presenting issues, and sociodemographic groups. Overall, findings demonstrated a change in service use at the onset of the pandemic in March 2020. Specific findings included changes in the number of contacts with Kooth per user, certain referral sources and specific presenting issues, but more limited changes regarding socio-demographics characteristics. Less consistent evidence emerged for a sustained change in service use one-year post-pandemic. These patterns are consistent with existing

**Table 6. Interrupted time-series model by deprivation quartiles.**

| IMD rank Deprivation[1] | Q1 | | Q2 | | Q3 | | Q4 | | Q5 | |
|---|---|---|---|---|---|---|---|---|---|---|
| | RR | 95% CI | RR | 95% CI | RR | 95% CI | RR | 95% CI | RR | 95% CI |
| Pre-intervention slope | 1.04 | 0.88, 1.23 | 1.61 | 0.82, 3.14 | 1.32 | 0.6, 2.87 | 1.01 | 0.99, 1.02 | 0.79 | 0.56, 1.12 |
| Change in intercept | 1.58 | 1.01, 2.47 | 1.53 | 0.91, 2.58 | 1.37 | 0.76, 2.46 | 1.65 | 0.93, 2.92 | 1.47 | 1.06, 2.04 |
| Change in slope | 1.00 | 0.99, 1.00 | 0.99 | 0.98, 1.00 | 1.00 | 0.98, 1.01 | 0.99 | 0.97, 1.01 | 1.06 | 0.93, 1.21 |

[1] IMD: Index of Multiple Deprivation; Q1: Most deprived–Q5: Less deprived

research demonstrating significant changes in mental health service utilisation in England at the beginning of the pandemic at the time of the first lockdown in March 2020 [42].

## 4.1. Changes in referrals and pathways to receive digital care

The Covid-19 pandemic and associated lockdown, school and healthcare restrictions had an observable impact on changes in referrals to the digital service in the UK. Patterns of access to Kooth also differed according to referral sources before, during, and after the pandemic. Wider research reports changes in patterns of referrals during the pandemic period [43], with services experiencing fewer referrals during the pandemic [44, 45]. In terms of referrals to Kooth, findings from the present analyses found a step change at the beginning of the pandemic period, with an increase in referrals via family/friends, care/social/charity, and social media/internet. Conversely, there was a significant decrease in referrals to Kooth via educational settings. These findings can be interpreted in light of lockdown restrictions meaning that CYP were not attending school [10]. Therefore, with schools closed, promotion of Kooth via schools was no longer possible.

The increase in referrals via social media and family/friends could also be reflective of changes in social patterns during lockdown. Research suggests that social media use increased during the pandemic [46] for reasons including social support and information seeking [47]. Furthermore, this was a period marked by reduced rate of referrals and disruptions to traditional face-to-face mental health services [8, 42]. Thus, more CYP would likely have been seeking alternative forms of mental health support and been offered such support via primary healthcare providers [48].

## 4.2. Changes in service access

The Covid-19 pandemic had a significant impact in terms of population mental health and mental health service utilisation [49, 50]. CYP were disproportionately affected [5] with a deterioration in mental health symptoms in early lockdowns [51]. Social isolation enforced by pandemic lockdowns was associated with higher rates of depression and anxiety in children and adolescents [7]. Thus, there is a need for effective and timely support for CYP at the beginning of major events such as global pandemics.

The present findings demonstrated that there was a significant increase in the number of CYP accessing an established digital mental health platform at the beginning of the pandemic in the UK. Digital mental healthcare, including apps and web platforms can make mental health and wellbeing support more accessible during major crises [48]. Despite an increase in nationwide mental health concerns, mental health services such as secondary mental health care and child and adolescent services, observed a reduction in referrals [42, 52, 53]. Reasons for this reduction include a reduced mental health workforce, self-isolation, and fears of contracting Covid-19 [54, 55]. Thus, it is likely that because of this, a greater number of CYP sought accessible mental health support such as Kooth.

It is noteworthy that the immediate increase in Kooth's service access was followed by a rate of engagement which was lower than what was observed before the start of the pandemic. Two reasons could explain this. Firstly, it is possible that following lockdown, CYP adjusted to the situation, and potentially experienced some emotional stabilisation [56]. Thus, there could be reduced demand for support. Conversely, this post-pandemic reduction also aligns with findings from other mental health services in the UK which also demonstrated a reduction of service utilisation following the Covid-19 pandemic [53, 57]. Furthermore, it is likely that there was an increase of demand for face-to-face mental health service post-lockdown, with figures suggesting some return to pre-pandemic activity [58]. Some research has suggested

that the ease in lockdown from May-July 2020 saw an increased number of referrals to mental health services and admissions vs. pre-lockdown [59]. Furthermore, given that Kooth's referral sources are integrated into place-based services (e.g., education, community, and healthcare services), where lockdown measures were enforced, this could mean that referrals over the duration of the pandemic slowed down.

Overall, findings from the present analyses and the wider state of mental health service over the Covid-19 pandemic suggest that three things happened. Firstly, significantly more CYP immediately accessed digital mental healthcare platforms such as Kooth to receive accessible support at the onset of the pandemic. Secondly, there was a signposting and referral shortage, meaning that as the pandemic went on, less CYPs heard about Kooth, thus, having an overall negative knock-on effect on services access. Lastly, the ease of lockdown measures and potential increase of CYP increasing face-to-face support could explain the reduction in users accessing Kooth.

## 4.3. Changes in user activity

User activity on the Kooth platform differs according to self-directed (i.e., through online forum-based activities) vs. practitioner-directed (e.g., user-practitioner interaction via chats and messages) activity. Users can interact in both types of activity on the platform.

There was a significant increase in users accessing practitioner-directed support at the beginning of the pandemic. Reduced access to and capacity of traditional mental health services over the pandemic [59] could explain this increase in CYP accessing human-mediated direct mental health support via Kooth. It is likely that CYP who would normally access 'traditional' mental health support via the NHS were redirected to accessible, digital mental health support such as Kooth. Indeed, research highlights the shift towards digital forms of mental healthcare during the Covid-19 pandemic to overcome face-to-face barriers brought on by lockdown restrictions [22, 48].

Conversely, findings suggest that whilst there was a steady increase in users accessing self-directed content on Kooth via online forum-based activities (e.g., discussions, comments, and posts), there was no significant step-change increase at the onset of the pandemic. These findings are inconsistent with research suggesting that community support is an important factor in moderating the mental health effects of catastrophic events [60]. Sharing collective experiences can help individuals process negative events, and sharing mental health experiences online is found to support recovery and reduce stress [43, 61]. Hence, it is surprising that no significant increase in self-directed activity was observed at the start of the pandemic.

However, we must consider that Kooth is a digitally enabled therapy service where trained counsellors are directly interacting with users and also monitoring and moderating self-directed user activities. Thus, there may be resource limitations making it difficult for practitioners to balance the management of practitioner-directed and user-directed service access. This could mean that demand might not have been matched by resourcing and thus, increases in access to practitioner-directed support were not sustained. Therefore, it is critical to find a solution for digital services to manage this demand, especially in times of global emergencies where there is a surge of demand for accessible mental health support [62]. The wider system should still consider the extra support required as digital services such as Kooth could be considered as part of the emergency strategy assessments to support mental health access during times of crises.

## 4.4. Changes in user mental health and wellbeing presentations

Trends in presenting issues were assessed across five categories: (1) external difficulties, (2) mental health, (3) physical health, (4) risk, and (5) suicidality. A significant step-change

increase was found for all except physical health during the first lockdown, with no evidence of a sustained increase following the pandemic lockdown. This is in keeping with research suggesting immediate increases in mental health difficulties across young people and adults at the onset of the initial Covid-19 UK lockdown [63, 64].

In contrast, research suggests a sharp decrease in CYP presenting with anxiety, depression, and self-harm to GPs in the UK [65, 66]. Significant disruptions to primary and secondary mental health services during Covid-19 lockdown restrictions [8, 42] may have resulted in CYP with presenting issues such as anxiety, depression, self-harm, and suicidality seeking accessible forms of support such as Kooth. Indeed, wider research indicates that despite this decrease in presentations to healthcare settings, there was increasing concern in suicidal and self-harm related behaviours in CYP during the Covid-19 lockdown [67]. This could be attributed to factors such isolation, loneliness, low social support, and unavailability of accessible healthcare which are suggested to increase the risk of such risk-related issues [66, 68]. Hence, the disparity in rates between primary/secondary healthcare compared to Kooth.

Furthermore, there was a particularly large increase of CYP presenting with risk-related issues, such as trauma and abuse, to the Kooth platform. Issues of risk, abuse, and safeguarding have been a particular concern in the UK throughout the lockdown periods [69, 70]. Crucially, this was a particular concern as typical avenues where a child at risk would be identified, such as schools [71], were restricted.

Overall, the trends of access to Kooth and the disparity in CYP presenting to primary and secondary healthcare versus Kooth is interesting. Patterns suggest that there was not a decrease in CYP with risk and self-harm/suicide-related issues, but an unmet need in the healthcare system during the Covid-19 lockdown. Anonymous and text-based services such as Kooth may have provided vulnerable CYP an alternative and accessible support system at a time when access to standard face-to-face support was restricted [3, 68].

## 4.5. Changes in access by sociodemographic characteristics

Analysing CYP access to Kooth across sociodemographic characteristics provided a nuanced picture of service access over the pandemic period.

**4.5.1. Gender.** Firstly, when looking at gender, only users who identified as female had a significant increase in access before, during, after the pandemic period. This contrasts to findings regarding those who identified as agender, gender fluid, or male, where there was no significant pattern in access. These patterns are consistent with literature suggesting that those who identify as female are more likely to seek help for mental health difficulties, relative to males [28, 29, 72] and those with gender diverse identities [73]. It is possible that this could be a function of self-stigma regarding mental health and help-seeking for both male and gender diverse CYP [73–77]. Mental health platforms such as Kooth must consider appropriate promotion strategies to engage gender groups who may experience higher self-stigma regarding help-seeking. This is especially pertinent in times of emergencies, such as the Covid-19 pandemic, where, for example, gender diverse youth reported significant mental health difficulties and a lack of social support [78].

**4.5.2. Ethnicity.** Exploring CYP access to Kooth across ethnic groups revealed a significant increase in access before, during and after Covid-19 for users from White ethnic backgrounds, whilst those from Black ethnic backgrounds only displayed a significant step-change increase at the onset of the pandemic. This contrasts with those from Mixed ethnic backgrounds who only displayed a significant increase prior to the Covid-19 pandemic, and those from Asian backgrounds who displayed no significant increase in access before, during, or after the pandemic period. This overall disparity in access to Kooth is consistent with wider research which highlights inequalities in representation of CYP from White versus ethnic

minority backgrounds in mental health services [26, 27], potentially as a function of stigma [79]. This is particularly pertinent during Covid-19 where individuals from ethnic minority backgrounds were disproportionately affected by mental health difficulties [80, 81].

However, the differences in access between different ethnic minority groups paints a more detailed picture. Critically, the lack of significant increase among Asian/Asian British can be interpreted in light of existing research which suggests differences in mental health-related stigma across ethnic minority groups [82]. Furthermore, research has found that, compared with individuals from Black ethnic backgrounds, individuals from Asian backgrounds reported higher levels of self-stigma in relation to mental health [83]. This could be due to different cultural norms and beliefs surrounding the conceptualisation of mental health and seeking support [84]. This is critical to examine as racial disparities in mental health distress likely increased during Covid-19 [80, 85]. Thus, accessible mental health services such as Kooth should consider these inequalities when considering engagement strategies, and to be in a place of readiness for emergencies such as pandemics. Indeed, Kooth has worked closely with organisations to build partnerships to ensure that the service meets the needs of users. However, further consideration of variables such as cultural norms and beliefs is essential to ensure culturally appropriate engagement across.

Analysis of access by deprivation indices revealed differences in access across the quartiles. Specifically, there was a significant step change increase at the beginning of the pandemic for both the lowest and highest quartiles. Findings regarding the lowest quartile are promising as research suggests that those from deprived areas reported significantly greater mental health difficulties across Covid-19 [86]. Whilst mental health symptoms of anxiety or depression were generally event-related and temporary [56, 86, 87], vulnerable populations such as those experiencing financial difficulties, continued to experience enduring distress. Amongst those more at risk are people from deprived areas, minorities or those with existing conditions [88]. It is possible that this significant step-change increase in access to Kooth was a function of engagement activities. Indeed, Kooth has made efforts to reach out to communities from marginalised CYP, including those from more deprived communities [89].

A significant step change increase was also observed for CYP from the highest quartile. There are several possible reasons for this. Firstly, research suggests that those from higher socioeconomic quartiles are more likely to have higher mental health literacy [90], which is linked to greater help-seeking [91, 92]. Secondly, it is possible that CYP from higher socioeconomic backgrounds were more likely to already be in the mental healthcare system. Those from higher socioeconomic backgrounds are more likely to access mental health support [93] and are more likely to utilise private healthcare [94]. As such these CYP may have been more likely to have been referred to Kooth by physicians.

However, these findings should be interpreted with caution because the CCG IMD represented 200,000 to 500,000 residents and did not provide a localised measure of deprivation. Thus, it is difficult to accurately ascertain the reach of Kooth across the different groups—particularly the lower quartiles. Nonetheless, these findings provide a brief insight into the importance of tailoring engagement strategies to demographics groups, and aiming to integrate into healthcare systems and communities at both a national and local level. Improving engagement strategies can ensure that user needs are met and access can be boosted and retained during and after times of need [48].

## 4.6. Study strengths and limitations

A key strength of this study is the quality of the data collected. Kooth is a live digital service operating nationally in the UK and therefore, data was collected in real-time and within real-

world settings. This allowed for a large volume of data to be collected, and the opportunity to routinely collect demographic, service engagement, and outcome data. The large sample size, in particular, provided the opportunity to gather valid insights in presenting issues of CYP across age, gender, and ethnic groups. Furthermore, the volume of data was further supplemented with the use of public data on deprivation. Thus, we were able to make detailed inferences in our observations, by observing access changes on presenting problems, platform activity and referral sources.

Despite these strengths, the study should be interpreted considering its limitations. Firstly, 50% of service users consented to their data being used. We compared sample distributions between consented and non-consented users and no large differences emerged in terms of demographic distribution. However, it is difficult to ascertain the extent to which this could have impacted key findings. Secondly, the service users were primarily White or identified as Female. This is an issue as mental health difficulties and access to support varied according to sociodemographic characteristics. For example, individuals from ethnic minority backgrounds experienced significant mental health distress during the pandemic [80]. Thus, it is possible that the present analysis did not accurately capture CYP needing support. Thirdly, whilst the use of the multiple deprivation index allowed us to draw conclusions regarding socioeconomic factors and Kooth access, certain limitations must be considered. The multiple deprivation index is an area-level measure and thus, this limits the interpretations of our findings due to its large scope at a CCG regional level. This is a particular issue for those from large and diverse areas, such as London, where patterns of deprivation are more complex. For example, larger populations from both higher and lower socioeconomic groups may be present within the same area. Thus, the indices may not accurately capture each group. Future research can, perhaps, compare different deprivation indices. The use of 0.05 significance levels, while widely used, has been challenged in particular when related to discovery of new effects [95]. Future studies could look to apply more stringent significant levels to strengthen the evidence. There is a possibility of the time Kooth commenced within in a CCG affecting the rate of change pre-pandemic (e.g. rate of change may be higher upon commencement then stabilise). We attempted to mitigate this by only including CCGs with a stable contract over the pre and post pandemic period. However, contracts that have been live longer prior to the pandemic were more established and may have differed from newer contracts.

## 4.7. Implications for future research

Findings from this study have clear implications in demonstrating the value of delivering accessible, digital mental health support for CYP—particularly in light of global crises such as the Covid-19 pandemic. However, future research must continue developing the evidence-base to understand user needs, and how digital mental health platforms such as Kooth can effectively meet the needs of CYP, and be effectively integrated into traditional mental health services. Firstly future research can supplement and extend these quantitative findings with a qualitative investigation to explore CYP and key stakeholder voices. Previous research has explored CYP experiences of accessing Kooth during the pandemic and has highlighted its value, particularly for vulnerable CYP [3]. However this, alongside qualitative investigations with CYP and mental health professionals [96, 97] highlight that future research must explore how a digital mental health platform such as Kooth can be integrated alongside face-to-face support (e.g., traditional counselling, school support, etc.) to complement, rather the replace, traditional services. Findings from the present analyses suggest that increased access to the platform was not sustained post-pandemic. Therefore, future qualitative research with CYP and key stakeholders, such as mental health professionals, can explore perspectives on the

value of how digital mental health can continue to provide valuable support for CYP in a post-pandemic landscape alongside other forms of support. Indeed, meaningful inclusion of CYP, especially those from marginalised backgrounds or with complex needs, has been highlighted as a key piece in increasing digital mental health uptake and integration [98].

Another avenue for future research is to extend findings from the present study and continue to examine access preferences for digital mental health interventions. In particular, attention should be given to exploring user behaviour and motivational factors influencing help-seeking behaviour. Studies should continue exploring what is hindering access to specific groups (e.g., across gender groups and ethnic minorities) and how digital technology and support services should be designed to promote equity of access to mental health [99]. In particular, future research can benefit from using different public reliable and available data repositories to give us a holistic understanding of access to digital mental health. Effectively using this data can help us to understand how digital interventions can contribute to positive outcomes in mental health and respond to a global crisis [100].

Referral patterns in the present analysis suggest that digital mental health platforms like Kooth could look to improve how the service is promoted via avenues such as social media and primary care settings. Reflections on the state of mental health care during and after the Covid-19 pandemic posit that research can seek to improve how digital mental health platforms are integrated into existing social and healthcare systems for effective referral pathways [101–103]. Given that referrals to Kooth via family/friends, care/social/charity, and social media/internet were not sustained following the pandemic, future research can seek to explore how effective promotion and integration of Kooth can be sustained longer-term. Such research is critical in the improvement of accessible mental health support for harder to reach communities and future major crises.

Furthermore, findings from the present analyses suggested differential access to Kooth across ethnic minority groups, and that referrals from word of mouth and friends increased during the pandemic. Given that mental health stigma is more common in some ethnic minority communities [83], further analysis of differences in referral sources between different ethnic groups before and during the pandemic could be valuable. Specifically, findings could provide valuable insights into whether the Covid-19 pandemic had an impact on mental health pathways for different cultural groups. Research into causes for low mental health access from ethnic minorities has raised mental health stigma as a key contributor [104, 105], which is likely to increase the importance of anonymity amongst Black CYP. Specifically analysing factors contributing to ethnic minority access to digital mental health support, alongside qualitative research with CYP, can provide a more nuanced and accurate understanding of this. Such research is beneficial in continuing to improve access to and promotion of mental health support to harder-to-reach communities such as CYP from different ethnic minority groups.

## 5. Conclusions

This paper revealed detailed patterns of CYP access to Kooth across the pandemic period. Overall, activity and contact with digital web-based therapy increased in the first few months of the pandemic, but, typically, no significant differences were found between pre and during the pandemic in service activity. These findings suggest that there was a greater need for support at the beginning of a crisis period, which in the future should be met with an increase in resources for services to help at the point of need. This is especially pertinent for those from more vulnerable or harder-to-reach populations, including those with existing risk-related issues and certain socio-demographic groups, such CYP from ethnic minority groups and lower socioeconomic status.

Overall, this research provides valuable data about service access and usage patterns of a digital platform such as Kooth and how it is used during a time of global need, particularly for vulnerable, harder-to-reach populations. Critically, findings provide an opportunity for organisational learning and research can be leveraged to inform and refine platform design and implementation within similar organisational settings. This is particularly relevant when tailoring such platforms for vulnerable, harder-to-reach populations. For example, there must be continued efforts to widen and tailor engagement strategies across key demographic groups and integrating across different referral pathways. Future research can also further explore barriers to help-seeking using digital mental health support across genders, different ethnicities, and discrete socio-economic indicators. This can ensure that digital mental health support effectively meets user needs, and inform engagement strategies to ensure that Kooth is reaching harder-to-reach populations. Such research can both facilitate engagement with digital mental health support overall, and for platforms such as Kooth to be in a place of readiness in future emergencies such as a global pandemic.

## Supporting information

**S1 Table. a.** The proportion of research consent responses relative to gender in the digital service. **b.** The proportion of research consent response relative to ethnicity in the digital service. (ZIP)

**S2 Table. Presenting concerns frequency and categorization for the study based on previous literature on Covid-19 mental health impact studies.**
(PDF)

## Acknowledgments

Prof. David Gunell to help put the team together, Charlotte Mindell and Crystal Oppong to support the project and secure the internal resources at Kooth Plc. Further thanks to Kooth analyst team Baptiste Duthoit, Cristina Gascón Garcia, Ellen Howard and Tom Kayll to support data extraction. Finally, to Luke Player and Yasmin Friedman to support the project administration in the ADP trusted research environment and the wider team at Swansea University.

## Author Contributions

**Conceptualization:** Duleeka Knipe, Santiago de Ossorno Garcia, Aaron Sefi, Ann John.

**Data curation:** Duleeka Knipe, Santiago de Ossorno Garcia, Louisa Salhi.

**Formal analysis:** Duleeka Knipe, Lily Mainstone-Cotton, Amanda Marchant.

**Investigation:** Duleeka Knipe, Louisa Salhi, Ann John.

**Methodology:** Duleeka Knipe, Santiago de Ossorno Garcia, Ann John.

**Project administration:** Santiago de Ossorno Garcia, Louisa Salhi.

**Resources:** Aaron Sefi.

**Supervision:** Aaron Sefi, Ann John.

**Validation:** Louisa Salhi, Lily Mainstone-Cotton, Ann John.

**Visualization:** Duleeka Knipe.

**Writing – original draft:** Duleeka Knipe, Santiago de Ossorno Garcia.

**Writing – review & editing:** Santiago de Ossorno Garcia, Louisa Salhi, Nimrah Afzal, Samaryah Sammut, Lily Mainstone-Cotton, Aaron Sefi, Amanda Marchant, Ann John.

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
