## [Decision Letter · Decision Letter 0]

20 Mar 2024

PONE-D-23-36010Digital Mental Health Service engagement changes during Covid-19 in children and young people across the UK: presenting concerns, service activity, and access by gender, ethnicity, and deprivationPLOS ONE

Dear Dr. Salhi,

Thank you for submitting your manuscript to PLOS ONE. After careful consideration, we feel that it has merit but does not fully meet PLOS ONE’s publication criteria as it currently stands. Therefore, we invite you to submit a revised version of the manuscript that addresses the points raised during the review process.

We look forward to receiving your revised manuscript.

Kind regards,

Giulia Ballarotto

Academic Editor

PLOS ONE

Journal Requirements:

"I have read the journal's policy and the authors of this manuscript have the following competing interests:

Dr Louisa Salhi and Aaron Sefi are currently employed by Kooth digital Health - the service that the service data was extracted from. Dr Louisa Salhi also holds honorary researcher status at the University of Kent.

Dr Santiago de Ossorno Gardia and Lily Mainstone-Cotton were previously employed by Kooth Digital Health at the time of data extraction but are no longer employed by Kooth.  

All other authors have no conflict of interest."

We note that one or more of the authors are employed by a commercial company: Kooth digital Health.

(2) Please also provide an updated Competing Interests Statement declaring this commercial affiliation along with any other relevant declarations relating to employment, consultancy, patents, products in development, or marketed products, etc.  

Within your Competing Interests Statement, please confirm that this commercial affiliation does not alter your adherence to all PLOS ONE policies on sharing data and materials by including the following statement: ""This does not alter our adherence to  PLOS ONE policies on sharing data and materials.” (as detailed online in our guide for authors http://journals.plos.org/plosone/s/competing-interests) . If this adherence statement is not accurate and  there are restrictions on sharing of data and/or materials, please state these. Please note that we cannot proceed with consideration of your article until this information has been declared.

6. We note you have included a table to which you do not refer in the text of your manuscript. Please ensure that you refer to Table 4 in your text; if accepted, production will need this reference to link the reader to the Table.

**Additional Editor Comments:**

I have seen the paper and the suggestions from the reviewers. In particular, I urge the authors to pay attention to the indications provided by Reviewer 2, who highlighted several critical points. Additionally, I encourage the authors to utilize a proofreading service, given the numerous errors in the text and the usage of the English language.

Reviewers' comments:

Reviewer's Responses to Questions

**Comments to the Author**

1. Is the manuscript technically sound, and do the data support the conclusions?

Reviewer #1: Yes

Reviewer #2: Partly

2. Has the statistical analysis been performed appropriately and rigorously? 

Reviewer #1: Yes

Reviewer #2: Yes

3. Have the authors made all data underlying the findings in their manuscript fully available?

Reviewer #1: No

Reviewer #2: No

4. Is the manuscript presented in an intelligible fashion and written in standard English?

Reviewer #1: No

Reviewer #2: No

5. Review Comments to the Author

Reviewer #1: This is a potentially valuable descriptive study addressing a topic of high policy priority. The study appears to have been appropriately conducted, but the write up needs a significant overhaul. There are multiple errors throughout the manuscript that suggest inadequate attention to copy editing prior to submission. These errors take the form of the inclusion of legacy placeholder text, incorrect referencing of a table/figure, nonsensical sentences and a disconnect between the number of studies referred to (multiple) and supporting citations (singular). Other potential improvements include: framing this more explicitly as an evaluation of changes observed in one digital mental health service rather than digital mental health services more generally (eg modify title, describe role of Kooth in broader ecosystem, discuss generalisability to digital mental health services more widely); shorten discussion and focus on both lessons for Kooth and UK/international digital mental health services; and add nuance to statements about attractiveness of Kooth type services by referencing the youth and digital mental health services preference literature.

Reviewer #2: This study investigates CYP service use before and during the COVID pandemic. It stratifies its findings by type of use, reason for accessing service and demographic variables. Overall service use increased rapidly at pandemic inception but then reduced, though remained higher than pre-pandemic levels, time-series analysis revealed overall small reduction in rate of change of use during pandemic.

Introduction

Can the intro better relate to the research questions more clearly, why is referral route an important research question to ask? How does RQ about presenting concerns relate to the first paragraph.

“. Some countries responded with a national digital support provision for mental health in the face of the pandemic10, but it is not yet clear how well these services were utilized, or whether access was equitable across sociodemographic groups.”

Is Kooth a UK “national digital support provision” - if so, probably should state in the introduction and explain a little bit about how CYP can access this service to orient the reader.

Methods

Outcome variable

“Service access contacts was our outcome of interest by month. To calculate the rate of contact per user, we identified the number of active users on the platform (i.e., denominator) by counting the number of users accessing the service at least once each month between June 2019 to April 2021.”

For clarity can you state that the numerator was the number of contacts, denominator number of active users providing a rate of contacts/user.

CCGs:

“A total of 34 regions structured by Clinical Commissioning Groups (CCG) with unchanged resource contracts were selected from a total of 97 (N=5 decreased; N=58 increased); the selection criteria of regions was used to prevent biases due to changes in resources that may affect the demand and capacity of the online service during the pandemic (resource increase or decrease during the observational period). The study selected only regions of the service that had the same allocated number of resources before and during the pandemic, remaining therefore constant during the period.”

You need to explain what a CCG is and how it fits into the UK healthcare system to a non-UK audience or any reader who is unaware of how services are commissioned in the UK.

What do you mean by resource allocation – resource allocated to Kooth, resource allocated to the CCG?

How did you know resource allocation had increased decreased? please state and cite where this information was obtained if relevant.

This will then make it clear why you selected only 34/97 CCGs, which shall reduce the number of users included in the analysis.

IMD

Typically, this is known as the index of multiple deprivation, no deprivations.

Can you clarify that no part of the user postcode was used to determine the IMD.

Can you clarify what you mean by “partially disclosed” when you state “We used the average IMD rank of each CCG location included (out of a possible 191(ranks?)) to calculate deprivation quintiles (higher ranks are least deprived) for each user in relevance to their area (partially disclosed at registration and group by CCG).”

Why are 29.4% of IMD data missing for CCGs, given this information is publicly available?

Gender

Can you describe for the reader what genders are included in the terms agender and gender fluid.

Presenting concerns

“A total of 118 reported different presenting concerns were identified for the study. Those were grouped into five high-level categories aligned with previous literature on the mental health impact of the pandemic and allowed for sufficient observations within each category to track trends over time.”

This information is presented later on

“The dataset presented 118 different types of presenting concerns, which were grouped into five categories representing the main mental health impacts of the Covid-19 pandemic for CYP identified in previous literature (Supplementary Table 2).”

Can you present this information only once and state what the categories are, which authors grouped them and how disagreements were resolved.

Analysis

“For this, we included a binary coded variable in the model which represented the pandemic period (i.e., model step change), as well as an interaction term between time and intervention which models a slope change. “

In the interaction term, what was the reference period?

General point:

Could the time Kooth commenced in a CCG affect rate of change pre-pandemic, for example rate of change might be higher upon commencement and then stabilise.

Results

The study would benefit from a description of how the sample were selected, the number excluded from CCG selection aswell as the number excluded from lack of consent and any other exclusion criteria. Can you also state the proportion of registered users who transferred to active users and therefore included in the analysis (some may register but never contact the service directly or indirectly).

We are also missing N’s, would help interpretation of the table in tables 3, 4 and 5 confidence intervals. For example in table 5 small Ns in Q2, Q3 and Q4 are indicated by wide intervals.

Supplementary Table 3 is missing from the supplementary.

Main findings:

“When comparing trends in the periods, before the pandemic there was evidence of a 25% (95% CI: 9%- 44%) increase rate in contacts per user per month (p=.001), this reduced to a 21% (95% CI: 4%-41%) increase per month during the pandemic” – missing a fullstop.

“this reduced to a 21% (95% CI: 4%-41%) increase per month during the pandemic”

Where is this statement reflected in table 1 and figure 1?

Overall, the figure shows a gradual increase in use of Kooth that sharply increases at the onset of the pandemic then gradually decreases, but does not return to pre-pandemic levels. The figure might benefit from a dashed line which shows what Kooth contacts would have looked like had the pre-pandemic slope continued. This may visually explain why the rate ratio of slope change in table 1 indicates a relative decrease in use (RR<1).

Similar questions about the same statistics for direct and indirect contacts.

Figure 2

Why in figure 2 do the pre-pandemic slopes by type of presenting concern indicate decreasing contacts per month, but in figure 1 when presented overall, indicate increasing contacts per month?

Are these sub-populations, if so this needs to be stated?

An explanation of external needs to be included in the main results and not just in the supplementary to aid the reader.

Needs to be reproduced with the X axis and the legend presented suitable for publication.

From the figure it appears there is a slope change in self-harm/suicidality that was non-significant in the model, is it worth mentioning this?

Again, lines to indicate what Kooth use would have looked like had the pre-pandemic slope continued may facilitate understanding of the table statistics.

Discussion

Overall, this section needs to be rewritten prior to publication. The results need to be summarised more clearly and concisely. As do the main implications of the findings.

Example Very long and incomplete sentence. “We found that service activity changed and increase during the pandemic but changes in the referral sources and the way people access to pathways of care more generally may have impacted the increase in service access and demand, as contacts per users were higher in pre-pandemic activity when compared, our data suggest that dramatic changes in the healthcare system are likely to impact too digital ecosystems of support, even when operating anonymously and with relatively ease of access online, interoperability between services may be important in order to .” Please be more specific here and link your interpretation to the specific result your draw your inference from.

Please refer to this paper Effects of the COVID-19 pandemic on primary care-recorded mental illness and self-harm episodes in the UK: a population-based cohort study (thelancet.com)

This examines trends in MI presentations by age and suggests that self-harm presentations were alarmingly low. Potentially the increased contacts seen in Kooth provided a resource for children and adolescents who were unable/unwilling to present to primary care. The paper also provides an example of how to visualise expected and observed trends, which the authors may find useful.

I am not sure we can draw firm conclusions from the CCG IMD analysis about the facility of Kooth to reach deprived populations - each CCG represents 200 to 500K residents, pops are much larger in London CCGs, within those areas deprivation will vary widely, without a more localised measure of deprivation it is difficult to be certain that Kooth did truly reach these groups using the data in this study. However, this is noted in the limitations.

Do you have any hypotheses as to why Asian and mixed CYP did not increase use at pandemic onset?

In the final conclusion paragraph the authors state “Whilst the cause for the decrease in access as the pandemic continued is not known” – I’d really like them to suggest some possible hypotheses for why this might be: what about pandemic apathy/ of users, pandemic apathy of CYP settings to promote Kooth, increasing resilience of CYP as initial crisis passed, or bias towards larger numbers of less active users each month (those registered for long periods of time) as time progressed?

6. PLOS authors have the option to publish the peer review history of their article (what does this mean?). If published, this will include your full peer review and any attached files.

Reviewer #1: **Yes: **Matthew Philip Hamilton

Reviewer #2: No

---

## [Author Response · Author response to Decision Letter 0]

25 Sep 2024

Thank you for your assistance and review of the manuscript “Digital Mental Health Service engagement changes during Covid-19 in children and young people across the UK: presenting concerns, service activity, and access by gender, ethnicity, and deprivation”. 

I am pleased to return the manuscript after major revisions, which address the reviewer's comments. We now feel that the manuscript is of higher quality and will fit the scope of PLOS ONE well. We would like to thank the editors for the flexibility in resubmission due to the maternity leave of our first author. 

We have added, as suggested, a revised Competing Interests section here as well as in the mauscript; 

“We note that one or more of the authors are employed by a commercial company: Kooth Digital Health. This does not alter our adherence to PLOS ONE policies on sharing data and materials. 

Dr Louisa Salhi, Dr Nimrah Afzal, Samaryah Sammut and Aaron Sefi are currently employed by Kooth Digital Health and receive honorarium for their time. Dr Louisa Salhi also holds honorary researcher status at the University of Kent and the University of Manchester. Dr Nimrah Afzal holds an honorary research position at the University of Bath. 

Dr Santiago de Ossorno Gardia and Lily Mainstone-Cotton were previously employed by Kooth Digital Health at the time of data extraction and received honorarium for their time, but are no longer employed by Kooth Digital Health. 

Duuleka Knipe, Amanda Marchant and Ann John declare no conflict of interest, no remuneration was received from this work.” 

We confirm that neither the manuscript nor any parts of its content are currently under consideration or published in another journal. All authors have approved the manuscript and agree with its revised submission to the PLOS ONE. For ease of reviewing our changes, we have provided direct responses to the reviewers below this cover letter as well as a marked-up manuscript copy and a clean copy. 

Best Regards,

Dr Louisa Salhi (Corresponding author)

---

## [Decision Letter · Decision Letter 1]

8 Nov 2024

PONE-D-23-36010R1Digital Mental Health Service engagement changes during Covid-19 in children and young people across the UK: presenting concerns, service activity, and access by gender, ethnicity, and deprivation

PLOS ONE

Dear Dr. Salhi,

Thank you for submitting your revised manuscript to PLOS ONE.

The manuscript has been evaluated by a new reviewer, and their comments are available below.

The reviewer has made a number of suggestions for further revisions to the manuscript. Please note that we do not consider any of the requests to be mandatory, especially those to add citations, but you may agree that they would improve the manuscript. Please also ensure that the main manuscript file does not contain a Data Availability Statement that differs from the previously agreed one in the submission form.

We look forward to receiving your revised manuscript.

Kind regards,

Patrick Goymer

Staff Editor

PLOS ONE

Journal Requirements:

Reviewers' comments:

Reviewer's Responses to Questions

**Comments to the Author**

1. If the authors have adequately addressed your comments raised in a previous round of review and you feel that this manuscript is now acceptable for publication, you may indicate that here to bypass the “Comments to the Author” section, enter your conflict of interest statement in the “Confidential to Editor” section, and submit your "Accept" recommendation.

Reviewer #3: (No Response)

2. Is the manuscript technically sound, and do the data support the conclusions?

Reviewer #3: Yes

3. Has the statistical analysis been performed appropriately and rigorously? 

Reviewer #3: I Don't Know

4. Have the authors made all data underlying the findings in their manuscript fully available?

Reviewer #3: No

5. Is the manuscript presented in an intelligible fashion and written in standard English?

Reviewer #3: Yes

6. Review Comments to the Author

Reviewer #3: I found this a well written, interesting and useful paper (as a researcher who has done some work in this area and a practitioner working with children and young people), and felt I learned something from reading it. I commend the authors’ efforts to make use of available routine clinical data for research purposes, and organizational learning (and the fact it support this learning should perhaps be acknowledged more). I appreciate also the manuscript has already been reviewed, at least once, and joining at this stage, there is the potential I offer quite a different opinion.

Overall, I thought the work reported was of good quality and ambitious in scope, thus warranting publication. Notwithstanding this overall appraisal, there were some areas I thought it could be further developed and list them below.

1. I think more detail could be provided about the type of research this is and references to support the approach taken, i.e., for me, it is a clinical data mining (CDM) study. So, the work of Epstein for example could be cited (see: Epstein, I. (2009). Clinical data-mining: Integrating practice and research. Oxford University Press.). There are also other sources that may be suitable, e.g., Davenhill, R., & Patrick, M. (Eds.). (1998). Rethinking clinical audit: The case of psychotherapy services in the NHS. Psychology Press.

2. One area I thought more could have been said, in the context of the introduction and discussion, was in relation to the voices of young people. I thought the extent to which this digitally based support may be preferable during the context of the pandemic (i.e., with physical distancing etc.), but not outside of it was glossed over, and there may be value in citing additional studies which addressed the ‘client’ or stakeholder perspective on digitally delivered support, and how this interlinks with other forms of support (see, e.g., Maddison, C., Wharrad, H., Archard, P. J., & O’Reilly, M. (2024). Exploring young people’s perspectives on digital technology and mental healthcare: pilot study findings. Mental Health Practice, 27(1).) Also, are there ways in which this client voice could be better integrated into this type of CDM study in future?

3. A few more minor issues. Consider how statistical significance is referred to on pp. 16-17 (see Benjamin, D. J., Berger, J. O., Johannesson, M., Nosek, B. A., Wagenmakers, E. J., Berk, R., ... & Johnson, V. E. (2018). Redefine statistical significance. Nature human behaviour, 2(1), 6-10). At the beginning of the discussion, consider wording used (The paper didn’t itself investigate, but rather the research reported within it did …). Towards the end of the paper, I’m unsure I agree with the authors that it showcases the value of the service, so much as provide valuable data about access to the service and how it is used. Here, the ‘local’ genesis of the research is perhaps worth acknowledging, i.e., it was undertaken in a single (large) organization and, as much as anything, supports organizational learning.

Beyond this, I think the point about data availability is perhaps something that the authors want to liaise with the editorial team about, as it appears the authors are working with an anonymized dataset and I’m unsure why this wouldn’t be made available to the public, for example, owing to a freedom of information request. Where Kooth as an organization sits in relation to FoI legislation is, for me, another matter, i.e., they are not a public authority, but they are funded by public monies, so, for me at least, there would be an argument to make the anonymized data available, at least in the interests of transparency for research practice.

7. PLOS authors have the option to publish the peer review history of their article (what does this mean?). If published, this will include your full peer review and any attached files.

Reviewer #3: No

---

## [Author Response · Author response to Decision Letter 1]

10 Dec 2024

Thank you for your response and for sending the reviewer comments of the manuscript “Digital Mental Health Service engagement changes during Covid-19 in children and young people across the UK: presenting concerns, service activity, and access by gender, ethnicity, and deprivation”.

I am pleased to return the manuscript after minor revisions addressing the reviewer's comments. Please see the responses to the reviewer’s comments below. 

1. I think more detail could be provided about the type of research this is and references to support the approach taken, i.e., for me, it is a clinical data mining (CDM) study. So, the work of Epstein for example could be cited (see: Epstein, I. (2009). Clinical data-mining: Integrating practice and research. Oxford University Press.). There are also other sources that may be suitable, e.g., Davenhill, R., & Patrick, M. (Eds.). (1998). Rethinking clinical audit: The case of psychotherapy services in the NHS. Psychology Press.

● Thank you for highlighting that the research approach could be emphasised. We have researched the clinical data mining approach and believe that this study does not fall under this approach, and neither is it a clinical data audit. This study is a hypothesis-driven time series analysis, which has been highlighted in the abstract, aims, and methods section. For example, in the aims on page 3: “We aimed to assess the impact of the pandemic on service use in terms of overall contact, referral source, type of contact, and the type of presenting issue (e.g. mental health, abuse) using interrupted time-series modelling, and explored whether the impact is different between sociodemographic groups (gender, ethnicity and deprivation level) in one digital mental health service for CYP operating in the UK.” and in the methods on page 4: “An interrupted time-series analysis was used to assess whether there was evidence of a change in online service use during the pandemic period (April 2020 - April 2021) compared to pre-pandemic (June 2019 - March 2020) trends.”

● However, we made some additions to further highlight the hypothesis-driven nature of this study in the Abstract - we hope that this makes the research approach more explicit. For example:

○ “The aim of this study was to assess whether service access and presenting concerns differed before and during the pandemic. Sociodemographic characteristics (gender, ethnicity, and deprivation level) were examined to identify disparities in service use.”

2. One area I thought more could have been said, in the context of the introduction and discussion, was in relation to the voices of young people. I thought the extent to which this digitally based support may be preferable during the context of the pandemic (i.e., with physical distancing etc.), but not outside of it was glossed over and there may be value in citing additional studies which addressed the ‘client’ or stakeholder perspective on digitally delivered support, and how this interlinks with other forms of support (see, e.g., Maddison, C., Wharrad, H., Archard, P. J., & O’Reilly, M. (2024). Exploring young people’s perspectives on digital technology and mental healthcare: pilot study findings. Mental Health Practice, 27(1).) Also, are there ways in which this client voice could be better integrated into this type of CDM study in future?

● Thank you for highlighting the importance of including more detail regarding the voices of young people and the contextual preference for digitally delivered support both during and outside of the pandemic. 

● In the Introduction, we have included further discussion on youth preferences for digital mental health - both during and outside of the pandemic - with particular emphasis on including qualitative research with young people. See addition on page 2: “One avenue for improving access to mental health support for CYP is utilising digital mental health support. There are numerous perceived benefits of online mental health interventions, such as anonymity, privacy, and emotional safety due to reduced emotional proximity to the practitioner, as well as increased flexibility, control, and accessibility to treatment (Bambling et al., 2008; Hollis et al., 2017; King et al., 2006). Research exploring youth perspectives suggests that CYP value the potential for interactivity, personalisation, privacy, and sense of anonymity of online support (Dallinger et al., 2022; King et al., 2008; Ludlow et al., 2023). The value of digital support was even more paramount during the Covid-19 pandemic, where digital platforms created opportunity for providing support amongst lockdown restrictions (Chavira et al., 2022; Rauschenberg et al., 2021), especially for CYP with existing vulnerabilities (Mindel et al., 2022). Digital mental health services, therefore, provided an avenue for providing accessible support during the Covid-19 pandemic.”

● In the Discussion, we have highlighted the need for further qualitative investigations to incorporate key stakeholder voices, particularly those of young people, to supplement the present quantitative findings. The need to explore how digitally delivered support can be interlinked with other forms of support has also been raised. Given that we believe that this is not a clinical data mining study, this has not been highlighted. But we hope that this addition highlights the need for future integration of client voice. We have also cited the Mindel et al., (2022) paper which is a qualitative study conducted with youth recruited via Kooth, the same setting as the present paper. Please see: 

○ Pp. 16-17: “Firstly future research can supplement and extend these quantitative findings with a qualitative investigation to explore CYP and key stakeholder voices. Previous research has explored CYP experiences of accessing Kooth during the pandemic and has highlighted its value, particularly for vulnerable CYP (Mindel et al., 2022). However this, alongside qualitative investigations with CYP and mental health professionals (Bell et al., 2024; Maddison et al., 2024) highlight that future research must explore how a digital mental health platform such as Kooth can be integrated alongside face-to-face support (e.g., traditional counselling, school support, etc.) to complement, rather the replace, traditional services. Findings from the present analyses suggest that increased access to the platform was not sustained post-pandemic. Therefore, future qualitative research with CYP and key stakeholders, such as mental health professionals, can explore perspectives on the value of how digital mental health can continue to provide valuable support for CYP in a post-pandemic landscape alongside other forms of support. Indeed, meaningful inclusion of CYP, especially those from marginalised backgrounds or with complex needs, has been highlighted as a key piece in increasing digital mental health uptake and integration (Stiles-Shields et al., 2023).. “

● P17: “Specifically analysing factors contributing to ethnic minority access to digital mental health support, alongside qualitative research with CYP, can provide a more nuanced and accurate understanding of this. “

3. A few more minor issues. 

Consider how statistical significance is referred to on pp. 16-17 (see Benjamin, D. J., Berger, J. O., Johannesson, M., Nosek, B. A., Wagenmakers, E. J., Berk, R., ... & Johnson, V. E. (2018). Redefine statistical significance. Nature human behaviour, 2(1), 6-10).

● Thank you for this comment. We have reviewed this citation with interest and agree it highlights an issue of importance. Given the nature of this study we feel that the 0.05 significance level we have chosen is appropriate as the recommendation for more stringent significant levels in the citation above is limited to new scientific discoveries. However, we agree that this raises a valuable point and have added to the limitations section of the paper: “The use of 0.05 significance levels, while widely used, has been challenged in particular when related to discovery of new effects (Benjamin et al., 2018). Future studies could look to apply more stringent significant levels to strengthen the evidence.”

At the beginning of the discussion, consider wording used (The paper didn’t itself investigate, but rather the research reported within it did …). 

● Thank you highlighting this, we have amended the wording so that this is more accurate. Please see page 12: “The research within this paper aimed to investigate the…”

Towards the end of the paper, I’m unsure I agree with the authors that it showcases the value of the service, so much as provides valuable data about access to the service and how it is used. Here, the ‘local’ genesis of the research is perhaps worth acknowledging, i.e., it was undertaken in a single (large) organization and, as much as anything, supports organizational learning.

● Thank you for highlighting this, we agree that this paper provides valuable data about service access and use, as well as opportunity for organisational learning. This has been highlighted in the Conclusion on Page 17. “Overall, this research provides valuable data about service access and usage patterns of a digital platform such as Kooth and how it is used during a time of global need, particularly for vulnerable, harder-to-reach populations. Critically, findings provide an opportunity for organisational learning and research can be leveraged to inform and refine platform design and implementation within similar organisational settings. This is particularly relevant when tailoring such platforms for vulnerable, harder-to-reach populations. For example…”

Beyond this, I think the point about data availability is perhaps something that the authors want to liaise with the editorial team about, as it appears the authors are working with an anonymized dataset and I’m unsure why this wouldn’t be made available to the public, for example, owing to a freedom of information request. Where Kooth as an organization sits in relation to FoI legislation is, for me, another matter, i.e., they are not a public authority, but they are funded by public monies, so, for me at least, there would be an argument to make the anonymized data available, at least in the interests of transparency for research practice.

● While we agree that there is value in making data publicly available where possible, due to the nature of this data we only make this data available for research studies dedicated to improving the service. Requests to access this data can be made through our research team and we have updated the Data Access Statement to clarify this. 

“The data analyzed in this study is subject to the following licenses/ restrictions: Under Kooth’s privacy policy, data can only be shared with trusted partners for research studies dedicated to improving the service. Requests to access these datasets should be directed to research@kooth.com”

---

## [Editor Report · Decision Letter 2]

12 Dec 2024

Digital Mental Health Service engagement changes during Covid-19 in children and young people across the UK: presenting concerns, service activity, and access by gender, ethnicity, and deprivation

PONE-D-23-36010R2

Dear Dr. Salhi,

We’re pleased to inform you that your manuscript has been judged scientifically suitable for publication and will be formally accepted for publication once it meets all outstanding technical requirements.

Kind regards,

Patrick Goymer

Staff Editor

PLOS ONE
---

## [Editor Report · Acceptance letter]

14 Jan 2025

PONE-D-23-36010R2 

PLOS ONE

Dear Dr. Salhi, 

I'm pleased to inform you that your manuscript has been deemed suitable for publication in PLOS ONE. Congratulations! Your manuscript is now being handed over to our production team.

Kind regards, 

on behalf of

Dr Patrick Goymer 

Staff Editor

PLOS ONE